# *Chrysanthemum morifolium* Flower Extract Inhibits Adipogenesis of 3T3-L1 Cells via AMPK/SIRT1 Pathway Activation

**DOI:** 10.3390/nu12092726

**Published:** 2020-09-06

**Authors:** Mak-Soon Lee, Yangha Kim

**Affiliations:** Department of Nutritional Science and Food Management, Ewha Womans University, 52 Ewhayeodae-gil, Seodaemun-gu, Seoul 03760, Korea; troph@hanmail.net

**Keywords:** *Chrysanthemum morifolium* flower, adipogenesis, AMPK, SIRT1

## Abstract

Chrysanthemum (*Chrysanthemum morifolium* Ramat) flowers (CF) are widely consumed as herbal tea in many countries, including China. The aim of the present study was to examine the anti-adipogenic effect of hot water extraction of CF (HCF) on 3T3-L1 cells and their underlying cellular mechanisms. HCF treatment inhibited lipid accumulation under conditions that did not show the toxicity of 3T3-L1 adipocytes. The activity of glycerol-3-phosphate dehydrogenase (GPDH), which plays an important role in glycerol lipid metabolism, was also reduced by HCF. Adipogenesis/lipogenesis-related mRNA expression levels of peroxisome proliferator-activated receptor-γ (PPAR-γ), CCAAT/enhancer-binding protein-α (CEBP-α), sterol regulatory element-binding protein-1c (SREBP-1c), fatty acid-binding protein 4 (FABP4), acetyl-CoA carboxylase 1 (ACC1), and fatty acid synthase (FAS) were suppressed by HCF in a dose-dependent manner. Moreover, HCF increased activities of AMP-activated protein kinase (AMPK) and sirtuin 1 (SIRT1), involved in lipid metabolism. These findings suggest that HCF inhibits adipocyte lipid accumulation through suppression of adipogenesis/lipogenesis-related gene expression and activation of the AMPK/SIRT1 pathway. Therefore, it suggests that HCF may be used as a potentially beneficial plant material for preventing obesity.

## 1. Introduction

Obesity results from the excessive accumulation of fat in the body due to an imbalance between energy intake and consumption. It is a chronic medical condition linked to an increased risk of type 2 diabetes, hypertension, arteriosclerosis, dyslipidemia, and various chronic diseases, and is recognized as a serious health problem worldwide [1]. Fat accumulation in the body occurs because of the increased number and size of fat cells during the differentiation of preadipocytes to adipocytes [2]. Adipocyte differentiation is a complex process that is controlled by three classes of transcription factors: peroxisome proliferator-activated receptors (PPARs), CCAAT/enhancer-binding proteins (CEBPs), and sterol regulatory element-binding proteins (SREBPs) [3]. PPAR-γ and CEBP-α are considered to play critical roles in the process of adipocyte differentiation [4]. Both regulate the expression of SREBP-1c, adipocyte fatty acid-binding protein 4 (FABP4), acetyl-CoA carboxylase-1 (ACC1), and fatty acid synthase (FAS), which stimulate lipid accumulation within the cells [5,6].

AMP-activated protein kinase (AMPK) is a fuel-sensing enzyme that is activated by an increased ratio of AMP/ATP [7]. The sirtuin family, including sirtuin 1 (SIRT1), which is the mammalian homolog of yeast silent information regulator 2, are nicotinamide adenine dinucleotide (NAD+)-dependent deacetylases that are also regarded as fuel-sensing enzymes [8]. AMPK and SIRT1 play key roles in regulating lipid metabolism to maintain cellular energy balance, and phosphorylate various target proteins involved in lipid metabolism [7,8]. AMPK-deficient mice fed a high-fat diet to induce obesity exhibited increased body weight and fat mass and decreased SIRT1 expression in adipose tissues [9,10]. Therefore, obesity is associated with dysregulation of the AMPK/SIRT1 pathway, which can be seen to play an important role in the development of obesity.

*Chrysanthemum morifolium*, a representative plant of the Asteraceae family, is a perennial plant that has long been used for edible or medicinal, as well as ornamental purposes, in China. It contains major components, such as luteolin, apigenin, and chlorogenic acid [11], which have been shown to exhibit anti-obesity effects by suppressing weight gain [12,13,14]. In particular, extracts of *C. morifolium* Ramat flower (CF) have demonstrated bioactive properties, including anti-oxidant, anti-diabetic, and hypolipidemic effects [11,15,16,17], but few studies have evaluated its anti-obesity effects. In this study, we aim to investigate the anti-adipogenic effect of the hot water extract of CF (HCF) in 3T3-L1 adipocytes, and to understand its underlying regulatory mechanisms by focusing on AMPK/SIRT1 activation.

## 2. Materials and Methods

### 2.1. HCF Preparation

Edible *C. morifolium* Ramat is cultivated in the east, middle, south and southwest of China. It is classified into “Boju”, “Chuju”, “Gongju”, “Hangju” and “Huaju” varieties, based on production location and variations in processing methods (Chinese Pharmacopoeia, 2015 edition) [18]. Among them, “Hangju”, originating from Tongxiang county, is vastly cultivated in Zhejiang province. The sample of capitulum of “Hangju” is harvested in Lanxi from late October into November. An HCF (KL-CTEP009-001) was kindly supplied by Amway Korea (Gangnam, Korea) and Amway China R&D (Shanghai, China). The flower (500 g) was extracted with hot water at 95 °C for 1 h, followed by cooling and filtering of the extract, then spray drying it into a powder.

### 2.2. Ultra-Performance Liquid Chromatography (UPLC) Analysis

For the quantitative analysis of the bioactive compound, HCF (0.25 g) was added to 25 mL of 70% methanol. The mixture was ultrasonicated for 40 min and then filtered through a nylon filter (0.45 µm; Whatman, Maidstone, UK). UPLC analysis was performed using an Acquity UPLC^®^ H-Class system (Waters Co., Milford, MA, USA). The HCF was separated on an Acquity UPLC BEH C18 column (2.1 × 100 mm, 1.7 µm; Waters Co.). The mobile phases consisted of acetonitrile (Solvent A) and 0.1% phosphoric acid in water (Solvent B). The gradient elution used was as follows: 0–4 min, 90–84% B; 4–19 min, 84–82% B; 19–20 min, 82–70% B; 20–25 min, 70% B. The detector wavelength was 348 nm and the flow rate was 0.2 mL/min. The sample injection volume was 1 μL and the column temperature was constant at 35 °C.

Mixed standard stock solutions of luteolin-7-*O*-glucoside, luteolin-7-*O*-glucuronide, apigenin-7-*O*-glucoside, chlorogenic acid, 1,5-Dicaffeoylquinic acid, and 3,5-Dicaffeoylquinic acid were prepared in methanol with the concentration of 0.502, 0.524, 0.523, 0.491, 0.502, and 0.502 mg/mL, respectively. Quantification was done by external standardization, using the respective standards. Linearity was evaluated by the injection of 6 different concentrations which covering the expected concentrations of the samples.

### 2.3. Cell Culture

Mouse 3T3-L1 preadipocytes (American Type Culture Collection, Manassas, VA, USA) were maintained in Dulbecco’s modified Eagle’s medium (DMEM) containing 10% (*v*/*v*) bovine calf serum, 100 U/mL penicillin, 100 μg/mL streptomycin, and 2 mM glutamine at 37 °C, 5% CO_2_ [19]. To differentiate 3T3-L1 preadipocytes into adipocytes, 2 days after the preadipocytes became confluent (day 0, D0), they were replaced with a differentiation-inducing medium. Cells were exposed for 2 days in a differentiation-inducing medium containing 0.5 mM isobutylmethylxanthine, 1 μM dexamethasone, and 5 μg/mL insulin (MDI) (D2). The cells were cultured for 2 days in a post-differentiation medium containing 5 μg/mL insulin (D4), and then cultured for 5 days in DMEM containing 10% fetal bovine serum (D9). The medium was treated with HCF for 7 days (from D2 to D9).

### 2.4. Cytotoxicity Assay

Cell viability was measured using a CCK-8 kit (Dojindo Laboratories, Kumamoto, Japan) according to the manufacturer’s instructions. The 3T3-L1 preadipocytes were treated with HCF at a concentration of 0, 0.001, 0.01, 0.1, 1, or 10 μg/mL, and incubated for 2, 5, or 7 days. Absorbance was determined using a microplate reader at 450 nm (Varioskan Flash, Thermo Scientific, Waltham, MA, USA).

### 2.5. Intracellular Lipid and Triglyceride (TG) Assay

Intracellular lipid and TG contents were determined by Oil Red O staining (Sigma–Aldrich, St. Louis, MO, USA) and using a commercial TG assay kit (EMBIEL Co., Ltd., Gunpo, Korea), respectively [19]. To normalize intracellular protein concentration, the protein content was analyzed using a bicinchoninic acid protein assay kit (Thermo Scientific, Pittsburgh, PA, USA).

### 2.6. Glycerol-3-Phosphate Dehydrogenase (GPDH) Activity

GPDH activity was measured using a GPDH activity assay kit (Takara, Kyoto, Japan) in accordance with the manufacturer’s instructions [19]. The 3T3-L1 adipocytes were incubated for 7 days with 0, 0.1, 0.5, or 1 μg/mL of HCF. GPDH activity was measured by monitoring the absorbance at 340 nm using a microplate reader and normalized to the cellular protein content.

### 2.7. Real-Time Quantitative Polymerase Chain Reaction (RT-qPCR)

RT-qPCR was measured as previously described [19]. Total RNA from the 3T3-L1 adipocytes was extracted using TRIzol reagent (GeneAll Biotechnology, Seoul, Korea), and the cDNA was synthesized from RNA extracted using M-MLV reverse transcriptase (Bioneer, Daejeon, Korea). Afterward, RT-qPCR was performed in a fluorescent thermal cycler (Rotor-Gene^TM^ 3000, Corbett Research, Mortlake, NSW, Australia) using Universal SYBR^®^ Green PCR Master Mix (Bioneer Co., Daejeon, Korea). The 2^−ΔΔCt^ method was used as a relative quantification strategy, and target gene expression was normalized using β-actin as an endogenous control. Primers used for RT-qPCR analysis are shown in Table 1.

### 2.8. AMPK and SIRT1 Activity

AMPK activity was performed using an AMPK kinase assay kit (CycLex, Nagano, Japan) and quantified by measuring absorbance at 450 nm using a microplate reader [19]. SIRT1 activity was measured using a nuclear extraction kit (Abcam, Cambridge, UK), and then the nuclear fraction was extracted from cells using the SIRT1 activity assay kit (Abcam, Cambridge, UK). Fluorescence intensity was detected at 340 nm excitation and 460 nm emission, respectively, using a microplate fluorescence reader.

### 2.9. Statistical Analysis

Statistical analysis was performed using SPSS software (version 26; IBM Corporation, Armonk, NY, USA). Results were expressed as mean ± standard error (SE) of three independent experiments. Significant differences among different treatment concentrations of HCF were analyzed using a one-way analysis of variance (ANOVA), followed by Tukey’s multiple comparison tests; *p* < 0.05 was considered statistically significant.

## 3. Results

### 3.1. Content of Polyphenolic Compounds in HCF

Polyphenolic compounds in HCF were determined by UPLC analysis (Figure 1). From the analyzed results, three flavonoids, namely, luteolin-7-*O*-glucoside, luteolin-7-*O*-glucuronide, and apigenin-7-*O*-glucoside, and three phenolic acids, including chlorogenic acid, 1,5-Dicaffeoylquinic acid and 3,5-Dicaffeoylquinic acid, were detected, as shown in Table 2. Among the flavonoids isolated from HCF, luteolin 7-*O*-glucoside, luteolin-7-*O*-glucuronide, and apigenin-7-*O*-glucoside were 567.40 ± 3.65, 533.73 ± 3.38, and 671.76 ± 3.58 mg per 100 g, respectively. Of the three phenolic acids isolated from HCF, chlorogenic acid, 1,5-Dicaffeoylquinic acid, and 3,5-Dicaffeoylquinic acid were 329.41 ± 2.20, 101.12 ± 0.69, and 427.21 ± 2.28 mg per 100 g, respectively.

### 3.2. Effect of HCF on 3T3-L1 Cell Viability

To investigate the effect of HCF on lipid accumulation, we first confirmed the cytotoxicity of HCF against 3T3-L1 cells. Cells were incubated for 2, 5, or 7 days by treatment with HCF at concentrations of 0 (control), 0.001, 0.01, 0.1, 1, or 10 μg/mL. Cytotoxicity was unaffected by 0, 0.001, 0.01, 0.1, and 1 μg/mL of HCF after incubation for 7 days. However, after incubation with 10 μg/mL HCF for 7 days, cell viability significantly decreased by 14.5% compared with the control. Therefore, a non-toxic range of concentrations below 1 μg/mL HCF was used for further experiments (Figure 2).

### 3.3. Effects of HCF on Lipid and TG Content

Next, we analyzed the effects of HCF on the accumulation of lipid droplets of 3T3-L1 cells. Intracellular lipid content was measured by Oil Red O staining. Cells were incubated for 2, 5, or 7 days with 0 (MDI-treated control) and 1 μg/mL HCF. A change in adipocyte differentiation was observed at 7 days (Figure 3A). The intracellular lipid content associated with HCF treatment was suppressed by 17.1% relative to MDI-treated control cells at 7 days, and there was no significant difference at 2 or 5 days (Figure 3B). To measure the intracellular TG content, cells were treated for 7 days with 0 (MDI-treated control), 0.1, 0.5, and 1 μg/mL HCF. The intracellular TG content in the presence of 1 μg/mL HCF was inhibited by 23.9% compared with MDI-treated control cells (Figure 3C).

### 3.4. Effect of HCF on GPDH Activity

GPDH is an enzyme involved in TG biosynthesis by converting dihydroxyacetone phosphate to glycerol-3-phosphate in adipocytes [20]. Therefore, we analyzed the GPDH activity to explain whether GPDH affects the decrease in TG production by HCF in 3T3-L1 adipocytes. Cells were treated with 0 (MDI—treated control), 0.1, 0.5, and 1 μg/mL HCF, respectively, for 7 days. The GPDH activity in the presence of 1 μg/mL HCF was reduced by 24.2% relative to the MDI—treated control (Figure 4).

### 3.5. Effects of HCF on Adipogenesis/Lipogenesis-Related Gene Expression

We investigated whether HCF affects the regulation of adipogenesis/lipogenesis-related gene expression of PPAR-γ, CEBP-α, SREBP-1c, FABP4, ACC1, and FAS in 3T3-L1 adipocytes. Cells were treated with 0 (MDI—treated control), 0.1, 0.5, and 1 μg/mL HCF, separately, for 7 days. HCF repressed the mRNA levels of PPAR-γ, CEBP-α, SREBP-1c, FABP4, ACC1, and FAS in a dose-dependent manner (Figure 5).

### 3.6. Effects of HCF on AMPK and SIRT1 Activity

In this study, we evaluated the effects of HCF on AMPK and SIRT1 activity in 3T3-L1 adipocytes. For AMPK activity, cells were treated with 0 (MDI—treated control) and 1 μg/mL HCF and incubated for 7 days and then treated for 24 h with 1 μM of compound C (Comp C) as an AMPK inhibitor. The AMPK activity was increased following HCF treatment with or without Comp C (by 2.1- or 1.5-fold, respectively) relative to the MDI—treated control cells (Figure 6A). For SIRT1 activity, cells were incubated for 7 days following treatment with 0 (MDI—treated control) and 1 μg/mL HCF and then treated with 1 mM nicotinamide (NAM) as a SIRT1 inhibitor for 24 h. The SIRT1 activity was increased following HCF treatment with or without NAM (by 2.7- or 1.6-fold, respectively) in comparison to the MDI—treated control cells (Figure 6B).

## 4. Discussion

Obesity is a state in which body fat is excessively accumulated and may cause social problems because of decreased quality of life and the development of chronic diseases. Many researchers have attempted to determine whether natural food materials are effective in preventing obesity by inhibiting adipogenesis [21,22,23,24]. Recent studies demonstrated the anti-obesity effects of extracts of *Chrysanthemum indium* and *Chrysanthemum zawadskii* in the Compositae family [25,26,27,28]. In particular, *C. indium* extracts reduced body weight gain and fat deposition, and improved serum lipid profiles in high fat diet (HFD)-fed obese mice [25,26]. *C. zawadskii* extracts inhibited adipogenesis by suppressing the expression of transcriptional factors genes in 3T3-L1 adipocytes [27,28]. *Chrysanthemum morifolium* is reported to exert a variety of bioactive effects associated with its major components, notably flavonoids and phenolic acids [15,16,17,29]. Nevertheless, information on the anti-obesity effect of *C. morifolium* remains insufficient. In this study, we evaluated the effect of HCF on 3T3-L1 adipocytes to understand the potential regulatory mechanisms of its anti-obesity effects.

First, we analyzed the polyphenolic compounds in HCF and identified three flavonoids and three phenolic acids. The flavonoids were luteolin-7-*O*-glucoside, luteolin-7-*O*-glucuronide, and apigenin-7-*O*-glucoside, and the phenolic acids were chlorogenic acid, 1,5-Dicaffeoylquinic acid, and 3,5-Dicaffeoylquinic acid. In a previous study, luteolin and apigenin exerted an inhibitory effect on lipid accumulation during 3T3-L1 adipocyte differentiation and reduced weight gain in obese mice [12,13,30,31]. Chlorogenic acid exhibited an anti-obesity effect by improving lipid metabolism in obese mice [14]. Coffee fruit extract containing chlorogenic acid and caffeoylquinic acid showed anti-adipogenic and lipolytic effects in 3T3-L1 adipocytes [32]. In this study, we found that HCF containing polyphenolic compounds reduces intracellular lipid and TG accumulation in 3T3-L1 adipocytes. Therefore, it is speculated that the inhibitory effect of HCF on fat accumulation might be linked to the polyphenol compounds contained in HCF.

The differentiation of preadipocytes into mature adipocytes is affected by hormones and growth factors, and the expression of genes involved in intracellular lipid accumulation is increased [4]. During this process, there is induced expression of PPAR-γ, CEBP-α, SREBP-1c, and other transcription factors, and their interactions increase the expression of FABP4, ACC1, and FAS, which are involved in fat synthesis [4,6]. In the present study, HCF significantly down-regulated PPAR-γ, CEBP-α, SREBP-1c, FABP4, ACC1, and FAS in 3T3-L1 adipocytes. Previously, the ethyl acetate fraction of *C. indium* inhibited lipid accumulation in 3T3-L1 cells and suppressed the expression of PPAR-γ and CEBP-α/β/δ in white adipose tissue of obese mice [25]. The ethanol extract of *C. indium* showed a dose-dependent suppression of PPAR-γ, CEBP-α, and FAS expression in white adipose tissue and liver of obese mice [26]. The methanol and ethanol extracts of *C. zawadskii* reduced lipid accumulation and down-regulated PPAR-γ, CEBP-α, and FAS in 3T3-L1 adipocytes [27,28]. Our results suggest that HCF may exhibit anti-obesity effects by suppressing the expression of genes involved in adipogenesis and lipogenesis.

AMPK and SIRT1 are major enzymes that regulate lipid metabolism, and their activation leads to beneficial effects on metabolic diseases, such as obesity and type 2 diabetes [33,34,35]. AMPK requires NAD+ metabolism and SIRT1 activity to regulate energy expenditure, and AMPK activation is required for the action of SIRT1 [33,36]. In particular, 5-aminoimidazole-4-carboxamide-1-β-D-ribofuranoside, an activator of AMPK, inhibits adipocyte differentiation by suppressing PPAR-γ, CEBP-α, and FABP4 expression in 3T3-L1 cells [37]. Over-expression of SIRT1 attenuates adipogenesis in 3T3-L1 adipocytes and represses PPAR-γ, CEBP-α, and FABP4 expression [38]. A recent report associated the anti-obesity effect of *C. indium* with AMPK activation in adipose tissue of obese mice [25]. However, whether *C. morifolium* affects the regulatory mechanism of SIRT1, as well as AMPK, in adipocytes has not yet been elucidated. In this study, we first found that HCF stimulates AMPK and SIRT1 activity in 3T3-L1 adipocytes. Therefore, it can be assumed that the anti-adipogenic effect of HCF may be partially related to the AMPK/SIRT1 pathway in adipocytes.

## 5. Conclusions

In conclusion, our results show that HCF induces an anti-adipogenic effect by inhibiting lipid accumulation. It seems that this effect of HCF may be partially mediated through suppression of adipogenesis/lipogenesis-related gene expression and activation of the AMPK/SIRT1 pathway in 3T3-L1 adipocytes (Figure 7). Based on the results of this study, it is believed that further studies on animal models are needed to determine the potential efficacy of HCF as a material for preventing obesity.

## Figures and Tables

**Figure 1 nutrients-12-02726-f001:**
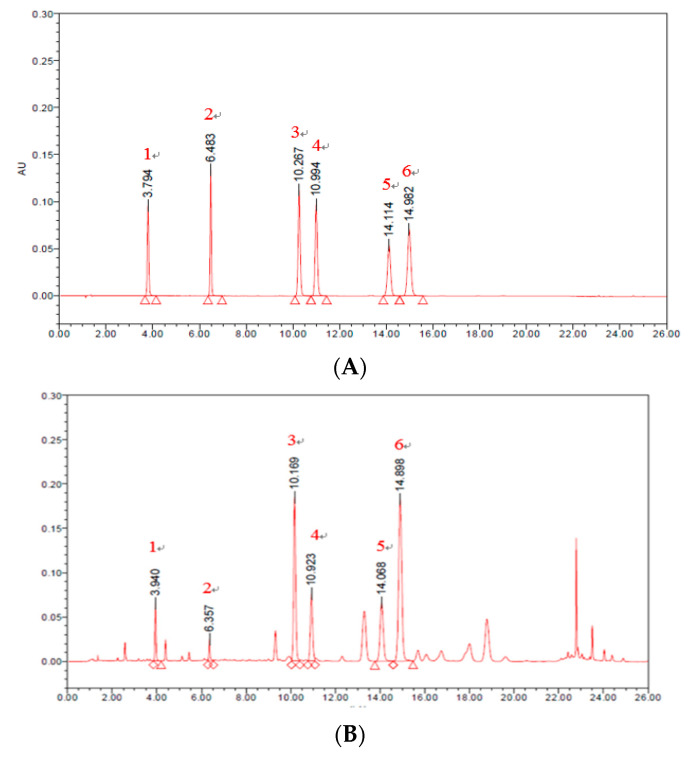
Ultra performance liquid chromatography (UPLC) chromatography of standard (**A**) and hot water extraction of Chrysanthemum (*Chrysanthemum morifolium* Ramat) flowers (HCF) (**B**). Peak 1, chlorogenic acid; Peak 2, 1,5-Dicaffeoylquinic acid; Peak 3, luteolin-7-*O*-glucoside; Peak 4, luteolin-*7-O*-glucuronide; Peak 5, 3,5-Dicaffeoylquinic acid; Peak 6, apigenin-*7-O*-glucoside.

**Figure 2 nutrients-12-02726-f002:**
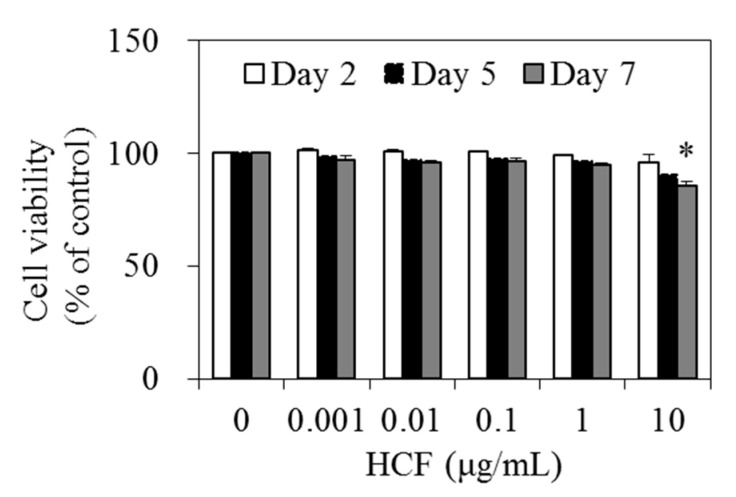
Effects of HCF on 3T3-L1 cell viability. Cell viability was determined using the water-soluble tetrazolium salt (WST)-8 assay. Cells were treated with 0 (control), 0.001, 0.01, 0.1, or 10 µg/mL HCF, and incubated for 2, 5, or 7 days. Values are expressed as a mean ± standard error of three independent experiments. * *p* < 0.05 vs. untreated control. HCF, hot water extract of *C.*
*morifolium* flower.

**Figure 3 nutrients-12-02726-f003:**
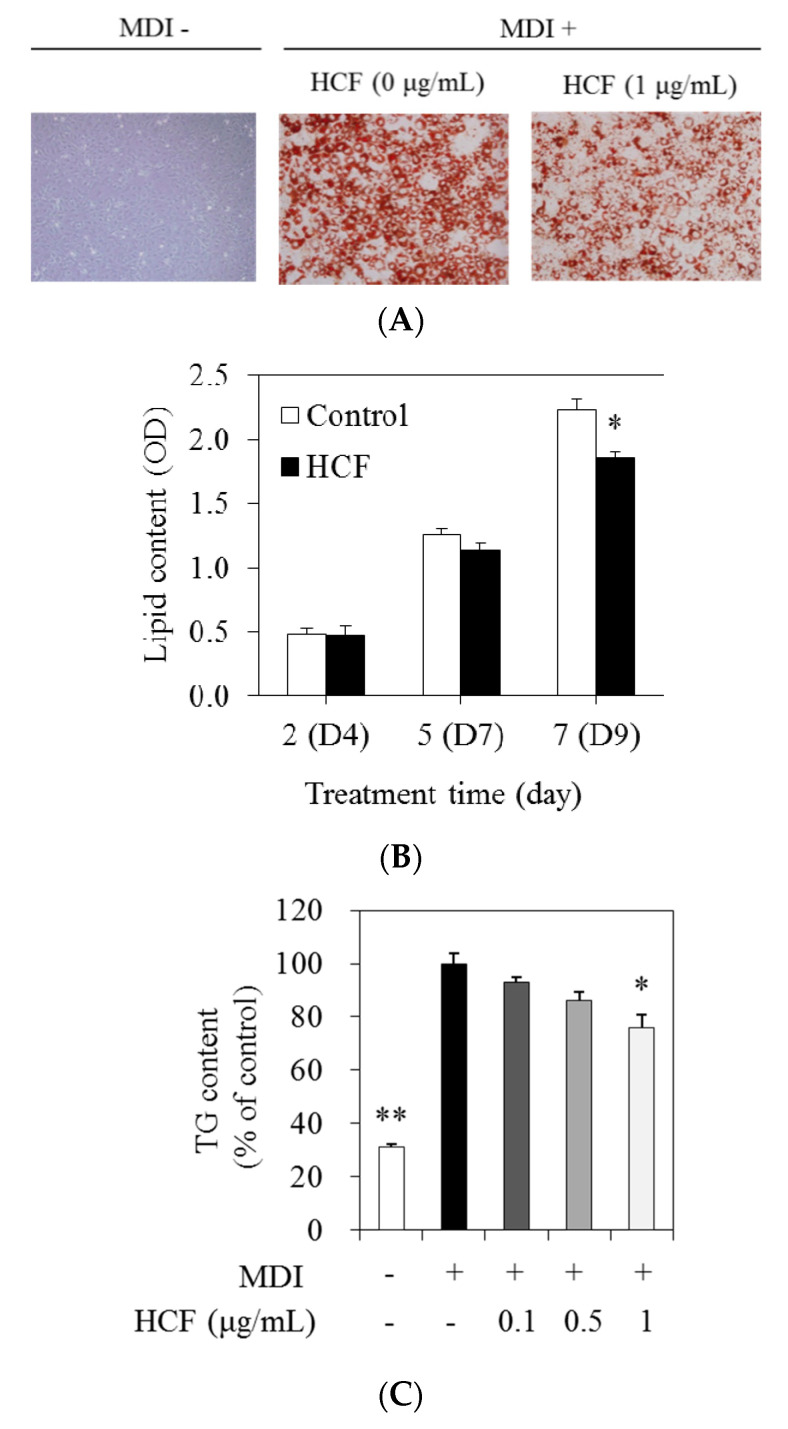
Effects of HCF on lipid accumulation and TG content during 3T3-L1 adipocyte differentiation. (**A**) The 3T3-L1 cells were treated with 0 (MDI—treated control) or 1 µg/mL HCF, and incubated for 2, 5, or 7 days (D2 to D9). On day 7 (D9), changes of adipocyte differentiation were observed by Oil Red O staining. Representative cell images were captured at 200× magnification. (**B**) Intracellular lipid content was stained with Oil Red O dye, and the stained oil droplets were dissolved with isopropanol and quantified by spectrophotometric analysis. (**C**) The 3T3-L1 cells were treated with 0 (MDI—treated control, MDI +), 0.1, 0.5, and 1 µg/mL HCF for 7 days. Preadipocytes without MDI media (MDI -) were used for baseline. The intracellular TG content was determined using enzymatic colorimetric methods. HCF, hot water extract of *C. morifolium* flower; TG, triglyceride; MDI, medium containing 3-Isobutyl-1-methylxanthine, dexamethasone, and insulin. Values are expressed as a mean ± standard error of three independent experiments. * *p* < 0.05 and ** *p* < 0.01 vs. MDI—treated control.

**Figure 4 nutrients-12-02726-f004:**
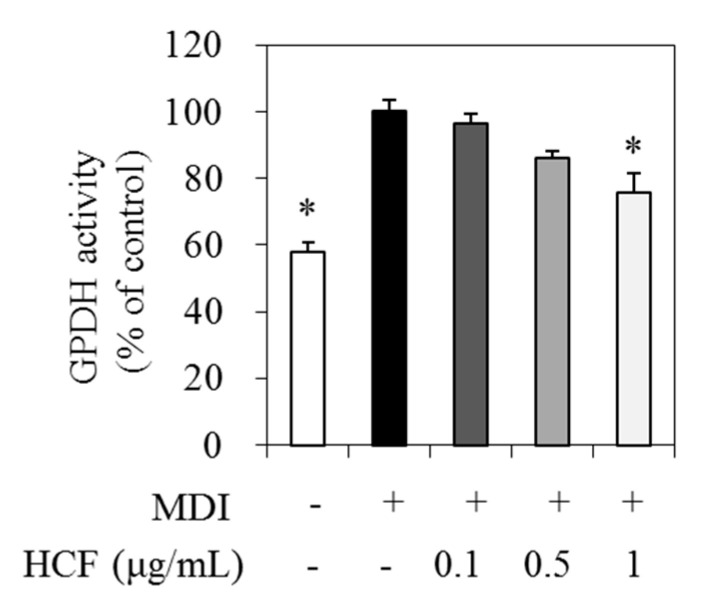
Effects of HCF on GPDH activity in adipocytes. The 3T3-L1 cells were treated with 0 (MDI—treated control, MDI +), 0.1, 0.5, or 1 µg/mL HCF for 7 days. Preadipocytes without MDI media (MDI -) were used for baseline. GPDH activity was determined using a GPDH assay kit. GPDH, glycerol-3-phosphate dehydrogenase; HCF, hot water extract of *C.*
*morifolium* flower; MDI, medium containing 3-Isobutyl-1-methylxanthine, dexamethasone, and insulin. Values are expressed as a mean ± standard error of three independent experiments. * *p* < 0.05 vs. MDI—treated control.

**Figure 5 nutrients-12-02726-f005:**
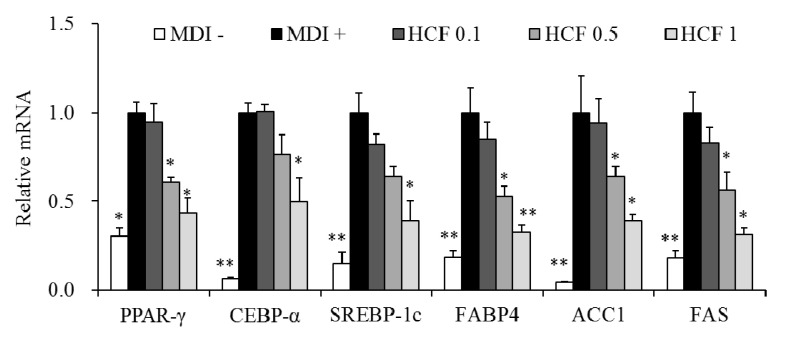
Effects of HCF on mRNA expression of adipocyte-specific genes in adipocytes. The 3T3-L1 cells were treated with 0 (MDI—treated control, MDI +), 0.1, 0.5 or 1 µg/mL HCF for 7 days. Preadipocytes without MDI media (MDI -) were used for baseline. The mRNA levels of PPAR-γ, CEBP-α, SREBP-1c, FABP4, ACC1, and FAS were determined by real-time quantitative PCR. HCF, hot water extract of *C.*
*morifolium* flower; PPAR-γ, peroxisome proliferator-activated receptor-gamma, CEBP-α, CCAAT/enhancer-binding protein-alpha, SREBP-1c, sterol regulatory element-binding protein-1c; FABP4, adipocyte fatty acid-binding protein 4; ACC1, acetyl-CoA carboxylase-1; FAS, fatty acid synthase; MDI, medium containing 3-isobutyl-1-methylxanthine, dexamethasone, and insulin. Values are expressed as mean ± standard error of three independent experiments. * *p* < 0.05 and ** *p* < 0.01 vs. MDI—treated control.

**Figure 6 nutrients-12-02726-f006:**
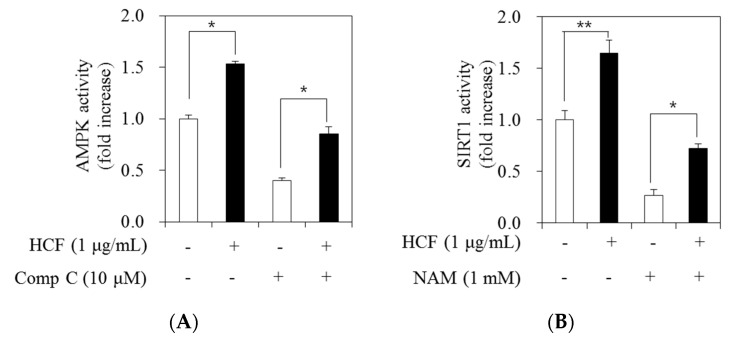
Effects of HCF on activities of AMPK and SIRT1 in 3T3-L1 cells. (**A**) The 3T3-L1 cells were treated with 0 ((MDI—treated control)) or 1 µg/mL HCF for 7 days, then incubated with 10 μM of Compound C (Comp C) as an AMPK inhibitor for 24 h. AMPK activity was determined using an AMPK kinase assay kit. (**B**) The 3T3-L1 cells were treated with 0 ((MDI—treated control)) or 1 µg/mL HCF for 7 days, and incubated with 1 mM nicotinamide (NAM) as a SIRT1 inhibitor for 24 h. SIRT1 activity was measured using a fluoromeric SIRT1 activity assay kit. HCF, hot water extract of *C.*
*morifolium* flower; AMPK, AMP-activated protein kinase; SIRT1, sirtuin 1. Values are expressed as mean ± standard error of three independent experiments. * *p* < 0.05 and ** *p* < 0.01 vs. untreated control.

**Figure 7 nutrients-12-02726-f007:**
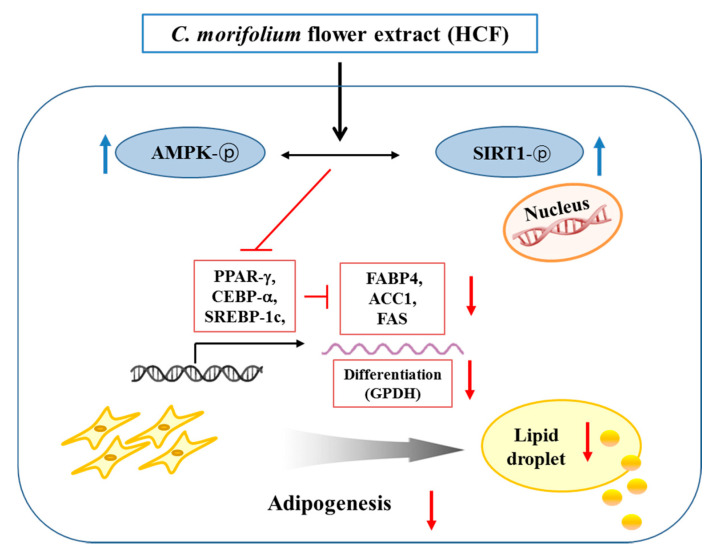
Schematic diagram showing potential regulatory mechanisms in adipogenesis by HCF in 3T3-L1 adipocytes.

**Table 1 nutrients-12-02726-t001:** Primers used for qPCR.

Name	GenBank No.	Primer Sequence (5′-3′)
ACC1	AY451393	F: CAAGTGCTCAAGTTTGGCGC
		R: CAAGAACCACCCCGAAGCTC
β-actin	NM_007393	F: GGACCTGACAGACTACCTCA
		R: GTTGCCAATAGTGATGACCT
CEBP-α	NM_007678	F: ATAGACATCAGCGCCTACAT
		R: TCCCGGGTAGTCAAAGTCAC
FABP4	NM_024406	F: CGACAGGAAGGTGAAGAGCA
		R: ATTCCACCACCAGCTTGTCA
FAS	AF127033	F: CTGGCATTCGTGATGGAGTC
		R: TGTTTCCCCTGAGCCATGTA
PPAR-γ	NM_011146	F: TTGATTTCTCCAGCATTTCT
		R: TGTTGTAGAGCTGGGTCTTT
SREBP-1c	AF509567	F: GGCTGTTGTCTACCATAAGC
		R: AGGAAGAAACGTGTCAAGAA

ACC1, acetyl-CoA carboxylase-1; CEBP-α, CCAAT/enhancer-binding protein-α; FABP4, fatty acid-binding protein 4; FAS, fatty acid synthase; PPAR-γ, proliferator-activated receptor-γ; SREBP-1c, sterol regulatory element-binding protein 1c.

**Table 2 nutrients-12-02726-t002:** Polyphenolic compounds identified in HCF.

Polyphenolic Compound	Content (mg/100 g)
Luteolin-7-*O*-glucoside	567.40 ± 3.65
Luteolin-7-*O*-glucuronide	533.73 ± 3.38
Apigenin-7-*O*-glucoside	671.76 ± 3.58
Chlorogenic acid	329.41 ± 2.20
1,5-Dicaffeoylquinic acid	101.12 ± 0.69
3,5-Dicaffeoylquinic acid	427.21 ± 2.28

Data are expressed as mean ± standard error of six replicates. HCF, hot water extract of *C. morifolium* flower.

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
