# Peer review of "Chrysanthemum morifolium Flower Extract Inhibits Adipogenesis of 3T3-L1 Cells via AMPK/SIRT1 Pathway Activation"

_nutrients, 2020, doi:10.3390/nu12092726_

Round 1

Reviewer 1 Report

The manuscript by Lee and Kim entitled: “Chrysanthemum morifolium Flower Extract Inhibits Adipogenesis of 3T3-L1 Cells via AMPK/SIRT1 Pathway Activation” presents a clearly written paper about mechanism behind the anti-obesity effect of the hot water extract of Chrysanthemum morifolium flowers in a 3T3-L1 cells model. The used references are up to date. The paper would be interesting for readers. 

Please find below my  comments:

As for the method and the origin of qRT-PCR conditions, was it published before or developed in the laboratory?

Please include chromatogram of C. morifolium extract analysis.

I would encourage the authors, when discussing Chrysanthemum anti-obesity effects, to discuss also the potential efficacy in animal model.

Author Response

Responses to Reviewer's comments (Reviewer 1)

The manuscript by Lee and Kim entitled: “Chrysanthemum morifolium Flower Extract Inhibits Adipogenesis of 3T3-L1 Cells via AMPK/SIRT1 Pathway Activation” presents a clearly written paper about mechanism behind the anti-obesity effect of the hot water extract of Chrysanthemum morifolium flowers in a 3T3-L1 cells model. The used references are up to date. The paper would be interesting for readers.

We thank the reviewer for careful reading and description about our manuscript with the valuable comments. We worked to the best of our abilities to revise the issues reviewer point out.

Please find below my comments:

As for the method and the origin of qRT-PCR conditions, was it published before or developed in the laboratory?

Response: We have added a previously published reference on qRT-PCR conditions (Page 3, lines 111).

Please include chromatogram of C. morifolium extract analysis.

Response: We added UPLC chromatography on Figure 1.

I would encourage the authors, when discussing Chrysanthemum anti-obesity effects, to discuss also the potential efficacy in animal model.

Response: We truly appreciate the reviewer's suggestion. We described in discussion on Page 8, line 248-249, and in conclusion on Page 9, line 294-295.

Reviewer 2 Report

Hello Authors 

The manuscript is well written and covers a novel applications of botanical extracts.

See my comments below,

1) Compare the HCF against a pharmaceutical anti-lipidemic active which is target specific.

Author Response

Responses to Reviewer's comments (Reviewer 2)

The manuscript is well written and covers a novel applications of botanical extracts.

We thank the reviewer for careful reading and description about our manuscript with the valuable comments.

See my comments below,

1) Compare the HCF against a pharmaceutical anti-lipidemic active which is target specific.

Thanks for the very helpful comment. We fully agree with the comments and concerns of the reviewers that HCF should be compared target-specific with a pharmaceutical anti-lipidemic active. This study originally aimed at investigating effect of HCF on adipogenesis during differentiation of preadipocytes into mature 3T3-L1 adipocytes. To test the functional mechanisms of anti-adipogenic effects of HCF, we evaluated the adipogenesis/lipogenesis-related gene expression, as well as AMPK/SIRT1 activity in 3T3-L1 adipocytes. In this study, we found some of the functional mechanisms of the anti-adipogenic effect of HCF were elucidated, but it is considered to be very limited. We highly appreciate the advice of reviewers that target-specific pathways for HCF should be compared using pharmaceutical anti-lipidemic active. Unfortunately, it was difficult to carry out this study right away because it was time consuming to prepare for further experiments. We will have to address this problem in the future research. Once again, thank you very much for your comment.

Reviewer 3 Report

Chrysanthemum (Chrysanthemum morifolium Ramat.) flowers (CF) are widely
consumed as herbal tea in many countries and this manuscript showed the obesity preventing effect of Chrysanthemum (Chrysanthemum morifolium Ramat.) flowers (CF)  herbal tea. This preliminary is interesting.

Author Response

Responses to Reviewer's comments (Reviewer 3)

Chrysanthemum (Chrysanthemum morifolium Ramat.) flowers (CF) are widely consumed as herbal tea in many countries and this manuscript showed the obesity preventing effect of Chrysanthemum (Chrysanthemum morifolium Ramat.) flowers (CF) herbal tea. This preliminary is interesting.

Dear reviewer,

Thank you very much for your favorable consideration.

Round 2

Reviewer 1 Report

Authors have addressed my all concerns.